# Influence of acute pain on valence rating of words

**Christoph Brodhun**[1], **Eleonora Borelli**[2], **Thomas Weiss**[1]*

**1** Department of Psychology, Clinical Psychology, Friedrich Schiller University, Jena, Germany,
**2** Department of Biomedical, Metabolic and Neural Sciences, University of Modena and Reggio Emilia, Modena, Italy

* thomas.weiss@uni-jena.de

**Data Availability Statement:** All data are given in the Ms and the Supplementary material.

**Funding:** The authors received no specific funding for this work.

## Abstract

Numerous studies showed the effect of negative affective and pain-related semantic primes enhancing the perceived intensity of successive painful stimuli. It remains unclear whether and how painful primes are able to influence semantic stimuli in a similar way. Therefore, we investigated the effects of noxious primes on the perception of the valence of subsequent semantic stimuli. In two experiments, 48 healthy subjects were asked to give their valence ratings regarding different semantic stimuli (pain-related, negative, positive, and neutral adjectives) after they were primed with noxious electrical stimuli of moderate intensity. Experiment 1 focused on the existence of the effect, experiment 2 focused on the length of the effect. Valence ratings of pain-related, negative, and positive words (not neutral words) became more negative after a painful electrical prime was applied in contrast to no prime. This effect was more pronounced for pain-related words compared to negative, pain-unrelated words. Furthermore, the priming effect continued to affect the valence ratings even some minutes after the painful priming had stopped. So, painful primes are influencing the perception of semantic stimuli as well as semantic primes are influencing the perception of painful stimuli.

## Introduction

There are numerous studies showing effects of different kinds of primes on the perception and processing of painful stimuli. Thus, the presentation of various types of stimuli with negative valence like pictures [1–5], films [6], sounds [7], and odors [8] before or during the application of painful stimuli increases pain ratings. A similar intensity-enhancing effect was shown repeatedly for pain-related words and words with negative valence [e.g. 9–12]. In contrast to neutral words, pain ratings were increased when pain-related words or words with negative valence had been shown before the application of painful stimuli. Overall, there is evidence suggesting that written words may be more suitable to elicit priming effects in contrast to pictures, spoken words or environmental sounds [13]. These results are in line with the assumptions of the motivational priming theory [14]. According to this theory, emotions can be seen as action dispositions. In this view, all emotions can be localized in a two-dimensional space, including the dimension of affective valence and the dimension of arousal (or arousal/activiation). The theory poses that defensive reflexes (including startle and reflex responses to pain) "increase in

**Competing interests:** The authors have declared that no competing interests exist.

amplitude when an orgnaism is aversively motivated" (p. 372); oppositely, defensive reflexes are reduced in amplitude when the subject is positively motivated. Therefore, increased pain ratings to painful target stimuli when primed by negative or pain-associated words are a consequence of negative emotional priming.

In addition to the motivational priming, several studies have shown a more pronounced priming effect towards higher pain ratings for pain-related primes compared to negative, but pain-unrelated primes. For example, the priming of a physically identical noxious stimulus by a pain-related adjective (e.g. excruciating) results in higher pain ratings than priming by a negative adjective (e.g. hostile) [9]. Note, that both adjectives produce a similar negative valence. Therefore, the higher pain perception reported to physically identical noxious stimuli primed by pain-related as compared to similarly negative, but non-pain-related adjectives cannot be explained by the motivational priming theory alone. However, such an effect was observed for both pictures [5,15] and words [9]. The theory of neural networks provides the theoretical background for this additional negative priming effect [16]. According to this theory, past pain experiences would lead to develop an associative memory network that can affect the processing of noxious stimuli at different levels. This pain network would strengthen its connections and increase its efficacy whenever we are exposed to real or potentially real painful stimuli, or also to stimuli that semantically represent harm or threat [12]. Thus, pain-related primes could activate specific pain-related neural networks (not merely negative valence) in contrast to negative, pain-unrelated primes. Consequential, pain-related primes lead to a more prominent increase of pain ratings [9,17,18].

It remains unclear whether an effect of painful primes exists on perception and processing of the valence of words. Until now, there is no study investigating such an effect. Therefore, the current study examines a possible effect of painful primes on perception and processing of the valence of words. According to the motivational priming theory mentioned above, we hypothesized that the negative emotional aspect of painful stimuli leads to more negative valence ratings of pain-related and negative, pain-unrelated target words in contrast to neutral words. With respect to the theory of neural networks, we additionally hypothesized that this effect would be more pronounced for pain-related target words compared to negative, pain-unrelated target words. This would support findings of a pain-specific priming effect in addition to a simple emotional priming effect. This hypothesis is in line with results from Ritter et al. [9] who showed a more pronounced priming effect of pain ratings for pain-related word primes compared to negative, pain-unrelated primes.

The current study aims primarily to show the effects of painful stimuli as primes for pain-related words as targets compared to negative, pain-unrelated targets. Valence ratings of pain-related words are expected to be more negative compared to negative, pain-unrelated words after priming with painful stimuli. To assess the effects, we conducted two experiments. In the first experiment we wanted to investigate whether an effect of painful primes on word valence rating exists. If such an effect could be observed, the question remains for how long the priming with painful stimuli continues to effect valence ratings of words. Previous studies have demonstrated long-lasting effects of different types of priming for hours or even a day, e.g. [19]. Consequently, the goal of the second experiment was to determine whether a possible effect lasts even without the painful stimulation and to identify possible moderating variables.

## Experiment 1

### Participants

Twenty-five volunteers (17 female and 8 male, 24.4 ± 3.7 years old) participated in experiment 1. All participants were healthy right-handed native German speaking university students

recruited via social media and mailing list. Written informed consent for participating according to the Declaration of Helsinki was obtained from all participants. The experiments were approved by the ethics committee of the Friedrich Schiller University Jena (vote No. FSV 14/ 04) prior to study commencement.

All subjects were free of acute or chronic pain according to a Likert scaled (0–10) life pain questionnaire and free of pain medication. A clinical interview and the additionally applied Beck depression inventory (BDI-2) [20] revealed no depressive symptoms in the subjects. Therefore, none of the participants was excluded due to a priori criteria of indices for chronic pain experiences or previous critical pain experiences or BDI-2 scores above 18. Subjects received a monetary reimbursement of 8.50 Euro per hour.

Sample size was calculated using a tool for a priori analyses [21]. Considering the fact that there are no studies investigating the effect of painful priming on the valence of words, we used previous results of our research group investigating the effect of affective priming on pain ratings as a starting point. Considering the number of measurements for each person we calculated a necessary sample size of approximately 20 participants with $\alpha = 0.05$ and power = 0.9.

## Pain stimuli

Both in experiment 1 and 2, a constant current stimulator (DS7H; Digitimer, Welwyn Garden City, UK) generated monophasic electrical stimuli for 3.5 s duration with a frequency of 200 Hz. These stimuli were applied to the tip of the middle finger of the left hand using the method of intracutaneous stimulation [11,22,23]. Therefore, an isolated golden pin electrode (diameter: 0.95 mm, length: 1 mm) was inserted into a small epidermal cavity of 1 mm diameter and about 1 mm depth and fixed with adhesive tape. This was done to reduce skin resistance to elicit a pain sensation. Care was taken to not cause any bleeding. A flexible stainless-steel electrode, fixed loosely around the first finger joint of the middle finger, served as reference electrode.

To determine the pain sensitivity of the participants, electrical stimuli were given according the following procedure. The subjects received an electrical stimulus and were asked to rate the stimulus after it had stopped. The intensity of stimulation was changed in accordance to the rating of the subjects. We used the method of limits to determine perception thresholds for somatosensory sensation and an intensity to evoke moderately painful perceptions (details below).

**Rating scale.**   The subjects were asked to rate each stimulus on a modified Ellermeier scale [24,25]. This scale consists of eight verbal categories. Each category is subdivided by numbers (0, no sensation; 1–10, just perceived but not painful; 11–20, clearly perceived but not painful; 21–30, very mildly painful; 31–40, mildly painful; 41–50, moderately painful; 51–60, strongly painful; 61–70, very strongly painful; for a more detailed description of the scale–see Ritter et al. [24]). The scale was presented on the same computer screen where the words and the valence scale were presented later in the experiment. The subjects were prompted to give just the numerical rating verbally.

**Somatosensory perception threshold.**   Starting with 0 mA, stimulation successively raised in steps of 0.05 mA until the subject reported a first sensation, i.e. the rating was >0 for the first time (usually not painful). Thereafter, stimulation was decreased by 0.05 mA until the subjects gave a pain rating of 0 again. Such increases and decreases were repeated four times around the threshold for a reported sensation. The last three intensities were used to determine the threshold.

**Intensity to evoke moderately painful perceptions.**   After the determination of the somatosensory perception threshold, the intensity was successively raised in steps of 0.5 mA

until the second threshold, fixed at a rating of 50, indicating the boundary of moderate pain to strongly painful pain, was determined. We used a rating of 50 as we expected some habituation in the course of the main experiment. We applied the same procedure as for the first threshold (increasing and decreasing intensity four times). The intensity threshold identified this way was used as the electrical stimulus in the experiment. A stimulus of moderate pain was used because it was shown that priming effects with electrical stimulation only occur for clearly painful and not for lower intensity stimulation slightly above the pain threshold [4,9,11].

## Word stimuli

We used 40 German adjectives as target stimuli in four categories– 10 pain-related (e.g. excruciating), 10 negative (e.g. hostile), 10 neutral (e.g. cubical), and 10 positive adjectives (e.g. exhilarating), as in previous studies [11,18]. Between categories, words were matched with respect to their word frequency and word length. Valence and arousal qualities were balanced for pain-related and negative words. Furthermore, arousal qualities of pain-related, negative, and positive words were balanced (for a more detailed description of the method–see Richter et al. [18]). Adjectives were shown in a pseudo-randomized order. We assured that not more than two consecutive words belonged to the same category to prevent summation effects.

## Study design

Experiment 1 was divided into four blocks of 100 trials. Each block lasted 14 minutes. The rest interval between blocks was not longer than 1 minute. During this time, subjects were able to drink a glass of sparkling water or just to relax. Taken together, the experiment proper performed in a special experimental noise-attenuating cabin (Industrial Acoustics Company GmbH, Niederkrüchten, Germany) took approximately 1 hour, while the whole experiment (including preparation, questionnaires, determination of the stimulus intensity) no longer than 2 hours.

**Structure of a trial.** In each trial, an adjective was presented on the screen for 500 ms for the participants to read (Fig 1).

The offset of the presentation of the adjective was followed by 2.5–3.5 s of black screen, which was followed by the presentation of the valence scale (Self-Assessment Manikin, 1 = most positive, 9 = most negative; see [26]) to assess the valence of the shown word. The subjects gave their answer verbally. All ratings of the subjects were entered on the keyboard by

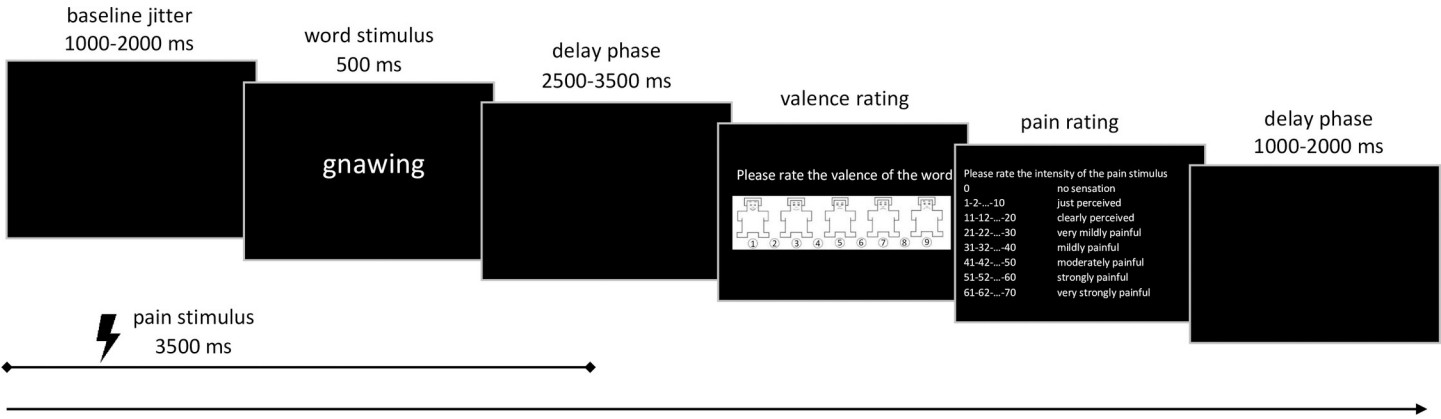

**Fig 1. Design of the experimental trials.** The used adjective and the scale were translated from German to English for illustration.

the experimenter. Additionally, on average every seventh trial participants were shown the modified Ellermeier scale [24,25] to rate the pain sensation verbally. Care was taken that every word category had the same number of pain ratings in the course of the experiment. At the end of the trial, there was a black screen for 1–2 s (second jitter) before the next trial started.

This primary structure of the trial was modulated by the presence or absence of an antecedent painful stimulation. Half of the trials did not have painful electrical stimulation before presentation of the adjective, the other half did. In the latter condition, the trial was as described above, just with additional electrical stimulation. Painful electrical stimulation started at the beginning of the trial, before an adjective was presented. Between one and two seconds (first jitter) after the onset of the electrical stimulation, an adjective was presented for 500 ms. The electrical stimulation lasted throughout the presentation of the adjective. The presentation of the scales was as described above.

In total, 200 trials with painful electrical stimulation and 200 trials without were conducted. The order of trials with and without painful electrical stimulation was randomised to avoid effects of expectations (for those effects–see [27]). So, each word was presented five times with painful electrical stimulation and five times without. Jitters were randomized and were applied to prevent a precise prediction of the onset of the next stimulus. Immediately after the last trial of the last block, subjects were given three minutes to recall any words shown during the experiment. The experiment was controlled by the software Presentation (Version 14.5, Neurobehavioral Systems, Inc., Albany, CA, USA).

## Data analysis

To test the effects of the pain stimuli both on the valence ratings and pain ratings, repeated measures analyses of variance (ANOVA) were applied. Within-subject factors were *Category* (pain-related, negative, neutral, and positive adjectives), *Word* (ten adjectives in each category), *Prime* (stimulation vs. no stimulation), and *Repetition* (first to fifth presentation of the adjective in the time course of experiment for both levels of factor *Prime*). Significant main effects were followed by post-hoc t-tests for paired samples according to our hypotheses (two-tailed, Bonferroni-Holm corrected). Normal distribution of all dependent variables was tested using the Shapiro-Wilk-Test. All variables met the assumption of normal distribution. In addition, the number of recalls after the experiment was also compared between word categories using an ANOVA for repeated measures. To account for violations of sphericity, the Greenhouse-Geisser procedure was used to correct degrees of freedom.

## Results of Experiment 1

ANOVA for valence ratings revealed significant main effects for *Prime* ($F_{(1; 24)} = 8.91$; $p = 0.006$; $\eta_p^2 = 0.27$), *Category* ($F_{(1.9; 46)} = 321.97$; $p < 0.001$; $\eta_p^2 = 0.93$), *Word* ($F_{(4.9; 118)} = 10.41$; $p < 0.001$; $\eta_p^2 = 0.30$), and *Repetition* ($F_{(1.5; 36.8)} = 7.61$; $p = 0.004$; $\eta_p^2 = 0.24$) as well as significant interactions for *Prime*\**Category* ($F_{(1.7; 39.9)} = 3.87$; $p = 0.036$; $\eta_p^2 = 0.14$), and *Category*\**Word* ($F_{(9.5; 227.3)} = 8.07$; $p < 0.001$; $\eta_p^2 = 0.25$). As the factor *Prime* only has two levels, the main effect showed a significant contrast stimulation vs. no stimulation, with more negative valence ratings (mean ± standard errors [S.E.]) of 5.39 ± 0.11 for stimulation vs. 5.14 ± 0.11 for no stimulation. Regarding the main effect for *Category*, t-tests between the levels of this factor revealed highly significant differences for all contrasts except for the contrast pain-related vs. negative words (t = -1.86; p = 0.076). The other contrasts for the main effect for *Category* (category with more negative valence ratings is mentioned first for each contrast): pain-related vs. neutral words (t = 14.54; p < 0.001), pain-related vs. positive words (t = 21.05; p < 0.001), negative vs. neutral words (t = 16.64; p < 0.001), negative vs. positive words

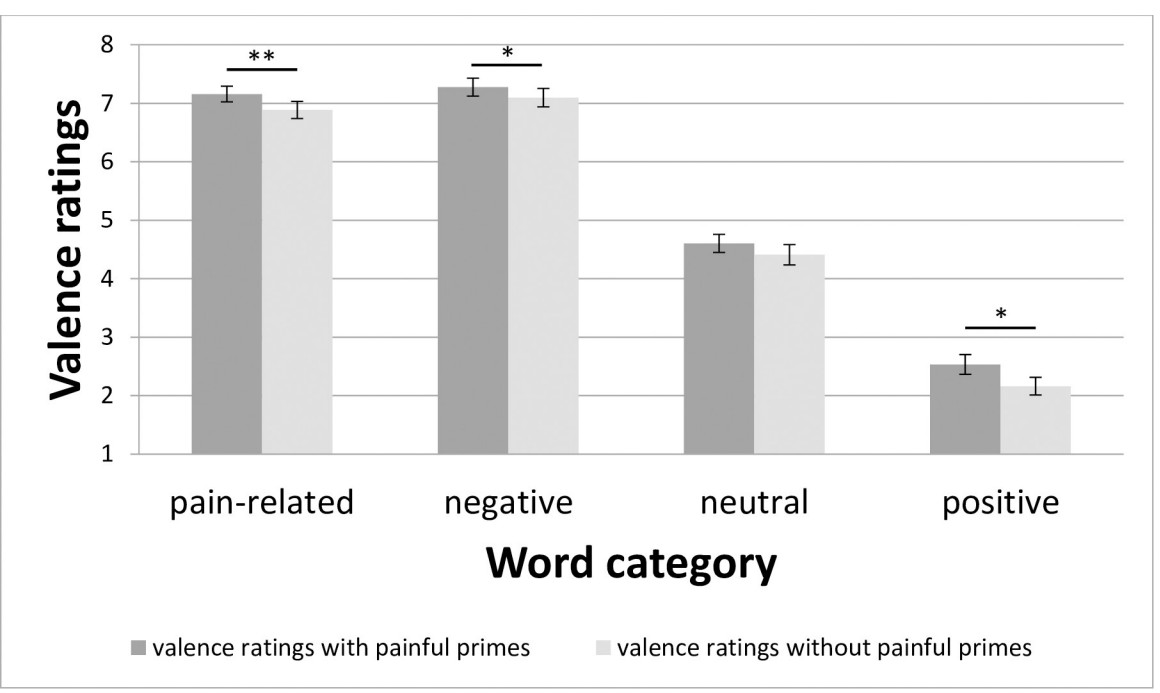

**Fig 2. Valence ratings with respect to the factors Prime and Category.** Data are reported as mean (SE); **: p < 0.01, *: p < 0.05.

(t = 21.32; p < 0.001), and neutral vs. positive words (t = 11.99; p < 0.001). Regarding the main effect for *Repetition*, t-tests between the levels of this factor revealed two significant differences. Repetition 1 showed lower valence ratings vs. repetition 4 (t = -3.74; p = 0.011) and vs. repetition 5 (t = -3.15; p = 0.039). The other contrasts for the main effect for *Repetition*: repetition 1 vs. repetition 2 (t = -2.44; p = 0.162), repetition 1 vs. repetition 3 (t = -2.81; p = 0.073), repetition 2 vs. repetition 3 (t = -2.33; p = 0.182), repetition 2 vs. repetition 4 (t = -1.46; p = 0.755), repetition 2 vs. repetition 5 (t = -0.71; p = 1), repetition 3 vs. repetition 4 (t = -0.13; p = 1), repetition 3 vs. repetition 5 (t = 0.36; p = 1), repetition 4 vs. repetition 5 (t = 1.33; p = 0.811). Regarding the main effect for *Word*, corrected t-tests between the levels of this factor revealed no significant differences for any contrast. Regarding the interaction effect *Category\*Word*, several significant contrast were found (post-hoc tests are presented in S2 Table in the supporting information). The interaction effect for *Prime\*Category* revealed significantly higher valence ratings with painful stimulation vs. no stimulation for pain-related (t = 3.92; p = 0.003), negative (t = 3.12; p = 0.014), and positive words (t = 2.98; p = 0.014), but not for neutral words (t = 1.82; p = 0.080) (Fig 2). To further investigate this effect, we computed difference variables, i.e. the difference between the valence ratings with pain stimulation and the valence ratings without pain stimulation and used these difference variables to compare the effects of pain stimulation between categories. ANOVA for this difference valence ratings revealed a significant effect ($F_{(3; 39.9)}$ = 3.87; p = 0.036; $\eta_p^2$ = 0.14). We found lower difference valence ratings for neutral vs. positive words (t = -3.12; p = 0.014), negative vs. positive words (t = -2.44; p = 0.046), and higher difference valence ratings for pain-related vs. negative words (t = 2.44; p = 0.046) (Fig 3). The contrasts pain-related vs. neutral words (t = 1.19; p = 0.355), pain-related vs. positive words (t = -1.21; p = 0.355), and negative vs. neutral words (t = -0.26; p = 0.400) were not significant.

ANOVA for pain ratings revealed significant main effects for *Prime* ($F_{(1; 24)}$ = 312.99; p < 0.001; $\eta_p^2$ = 0.93) and *Category* ($F_{(3; 72)}$ = 4.39; p = 0.007; $\eta_p^2$ = 0.15), but no interaction.

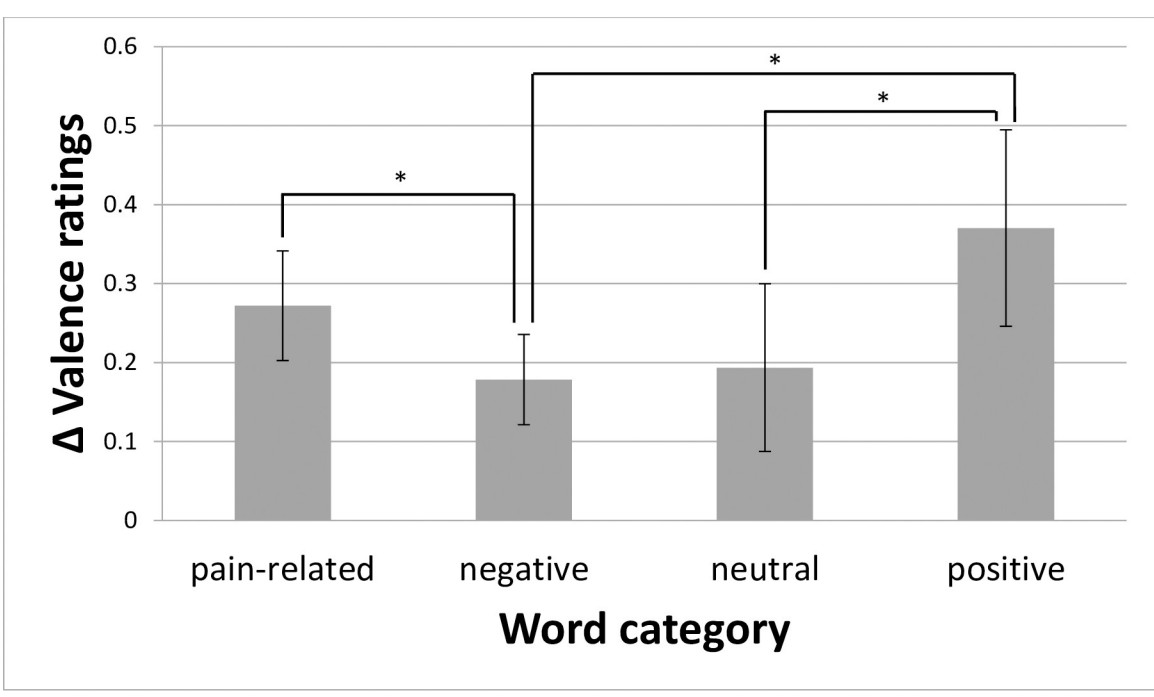

**Fig 3. Differences in valence ratings (Δ valence ratings) with respect to the assessed words (factor Category).** Data shown as differences of valence ratings for words with painful primes vs. the same words without painful primes. Data are reported as mean (SE); *: p < 0.05.

As the factor *Prime* only has two levels, the main effect showed a significant contrast stimulation vs. no stimulation, with higher pain ratings (mean intensity ± S.E.) of 48.10 ± 2.53 for stimulation vs. 4.17 ± 0.99 for no stimulation. Regarding the main effect for *Category*, t-tests between the levels of this factor revealed only one significant contrast with higher pain ratings for pain-related vs. neutral words (t = 3.93; p = 0.015) (Fig 4). The other contrasts for the main effect for *Category*: pain-related vs. negative words (t = 2.52; p = 0.093), pain-related vs. positive words (t = 2.20; p = 0.152), negative vs. neutral words (t = 1.18; p = 0.498), negative vs. positive words (t = 0.13; p = 0.896), and neutral vs. positive words (t = -1.72; p = 0.292).

ANOVA of the number of recalls per word category after the experiment showed a significant effect ($F_{(3; 72)}$ = 18.45; p < 0.001; $\eta_p^2$ = 0.43). Only the contrasts of pain-related words (mean recalled words (± S.E.) of 2.8 ± 0.36) vs. negative (5.1 ± 0.33) (t = -4.56; p < 0.001), neutral (4.76 ± 0.31) (t = -4.02; p = 0.002), and positive words (5.76 ± 0.27) (t = -7.78; p < 0.001) were significant with fewer recalls for pain-related words than all other categories. The other contrasts for this effect: negative vs. neutral words (t = 1.04; p = 0.308), negative vs. positive words (t = -1.83; p = 0.161), and neutral vs. positive words (t = -2.40; p = 0.073).

## Discussion to Experiment 1

We will mainly discuss the results of Experiment 1 with respect to aim of the study and to the two major theories described in the introduction. Several other aspects will be considered in the general discussion.

The main effect for *Category* revealed highly significant results for all contrasts of this factor except for the contrast pain-related vs. negative words. This result can be seen as manipulation check. It demonstrates the successful choice of stimulus material as in previous research [9,11,17,18].

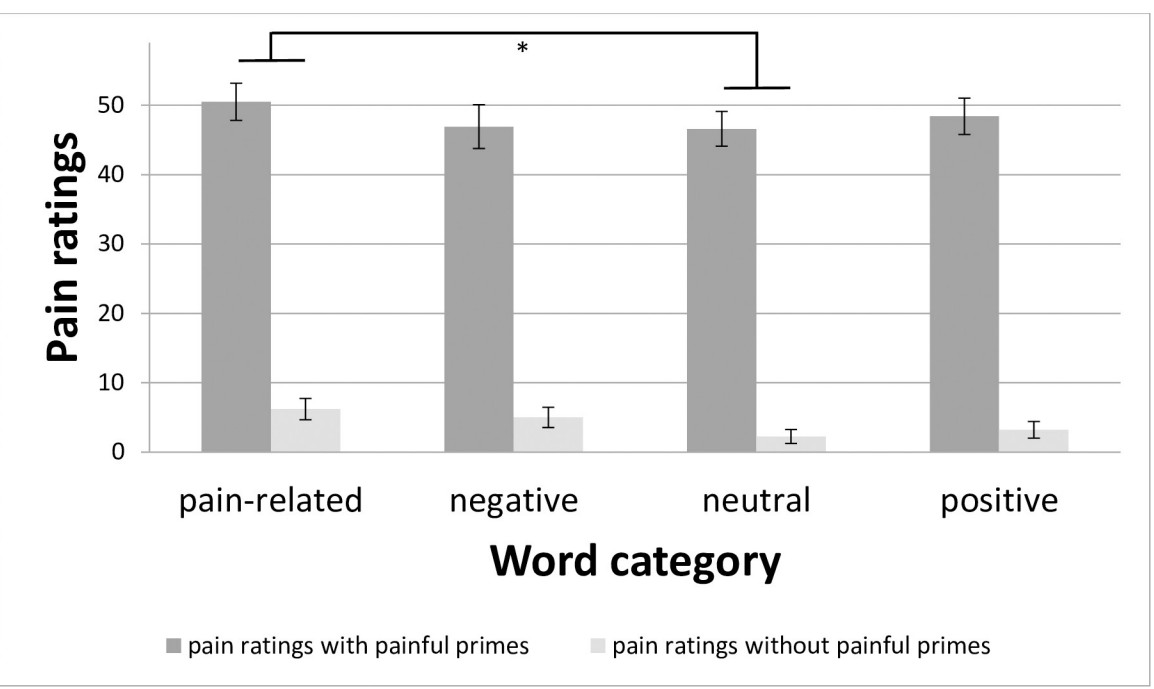

**Fig 4. Pain ratings with respect to the factors Prime and Category.** Data are reported as mean (SE); *: p < 0.05.

Valence ratings of words became more negative after a painful electrical prime was applied in contrast to no prime (Fig 2, main effect of *Prime*). This corresponds to the motivational priming theory [14]. The applied primes might lead to an activation of neural structures associated with negative affect. These results are in accordance with experimental findings supporting the assumptions of the motivational priming theory [e.g. 10,11]. The current study is the first to show this effect using painful electrical primes.

Results depicted in Fig 3 and the interaction *Prime*Category* indicate that painful primes do have a larger effect on valence ratings to succeeding pain-related words compared to negative, pain-unrelated words. This means that painful primes do not only lead to a simple emotional priming effect as predicted from emotional priming theory [14]. According to this theory, one would have expected a similar priming effect due to similar pain intensity and similar negative valence of the target words. However, we found an additional priming effect of noxious stimulation to pain-related adjectives; i.e., painful primes do not only lead to a more general negative valence rating, but to an exceedingly negative valence rating when followed by pain-related targets in contrast to negative, pain-unrelated targets. This result represents the answer to the major question of this investigation. The observed additional priming effect could be explained by the theory of neural networks [16]. This result is also in line with previous findings presenting higher pain ratings to physically identical noxious stimuli when pain-related adjectives served as primes in comparison to similarly negative adjectives [9].

No significant interaction between the factors *Word* and *Prime* was found. This indicates that the above-mentioned effect of the factors *Prime* and *Category* on valence ratings are not just due to a few items generating a strong effect. It rather shows a more general effect not just of individual words, but of word categories.

We found a main effect for *Repetition*. Further analysis revealed only two significant contrasts. Valence ratings were higher in repetition 4 and repetition 5 in contrast to repetition 1. This is an unexpected finding. It is not clear why this effect occurs only during repetitions 4 and 5.

Regarding pain ratings, we found higher ratings for trials with painful stimulation vs. no stimulation (main effect *Prime*). This means the manipulation with painful stimuli worked. Concerning factor *Category*, only the contrast pain-related vs. neutral words was statistically significant. So, for pain ratings we did not find a more pronounced effect for pain-related words compared to negative, pain-unrelated words (Fig 4), as described by Ritter et al. [9]. This may be because we did not specifically investigate the effects on pain ratings and, therefore, asked the participants merely in 60 out of 400 trials for a pain rating. So, the estimators may not have been robust enough. Subsequent power analyses for pain ratings with the found effect size revealed that additional 34 participants would have sufficed to reach a statistically significant contrast of pain-related words compared to negative, pain-unrelated words. Pain stimuli served as the prime, and pain ratings were not the target in our experiments. This also may have led to the absence of the effect as the effect might also be shorter-lasting. To analyse this aspect, further experiments are necessary.

## Experiment 2

### Participants, word stimuli, study design

Twenty-six volunteers (14 female and 12 male, 24.2 ± 4.5 years old) participated in experiment 2.

Generally, the procedure of experiment 2 was the same as in experiment 1 including similar written informed consent, intensity of pain stimuli used, and words used in the experiment. We also used the same scales as in experiment 1 (Fig 1). However, in addition to the 400 trials of Exp. 1 mentioned above, a pre and a post measure of the valence rating was added, each consisting of 80 trials (each of the words mentioned above was shown two times before and after the main experiment). The order of the words was pseudo-randomized as described above. In these two additional blocks, no painful electrical stimulation was applied. The pre-block served as a baseline for the valence rating, whereas the post-block served as a measure for the period the effect lasted. The post-block was split in half for analysis to examine the length of the effect (first 40 words in post-block 1 and the following 40 words in post-block 2). Each post-block lasted approximately six minutes.

In total, three subjects were excluded in experiment 2. One participant (female) did not reach the threshold for experiencing pain although the stimulator was at maximum. Two subjects (female and male) sensitised too much, the pain became too strong and the experiments were stopped by the experimenter because of the a priori criterion of a pain rating exceeding 70 on the modified Ellermeier scale. So, a total of twenty-three subjects was analysed in experiment 2 (12 female and 11 male, 24.6 ± 4.4 years old).

### Measures in Experiment 2

In addition to the above-mentioned recall questionnaire at the end of the experiment, several other questionnaires known to possibly influence pain processing were completed to identify possible moderating variables. The activation and inhibition system was assessed with the Behavioral Approach/Inhibition System Scale [28] because it is thought to play an important role in pain processing in both healthy participants and chronic pain patients [29,30]. Given the wealth of studies attesting a role of empathy in elaborating pain-related information (for an overview, see [31]), we also assessed the individual dispositions for empathy with the Interpersonal Reactivity Index [32]. Moreover, given the subjective nature of pain perception, individual differences in pain anxiety and catastrophizing were assessed with the Pain Anxiety Symptoms Scale [33] and the Pain Catastrophizing Scale [34].

## Data analysis

Data analysis was performed with an ANCOVA with similar factors as in Experiment 1 (within-subject factors *Category*, *Word*, *Prime*, and *Repetition*; however, the effects on the valence ratings were additionally examined by the within-subject factor *Block* (pre-block, main experiment, post-block 1, and post-block 2). Significant main effects were followed by post-hoc t-tests for paired samples according to our hypotheses (two-tailed, Bonferroni-Holm corrected).

As we have demonstrated effects in Exp. 1, we also used some parameters that sometimes have effects on pain ratings to exclude their influence on our results. Therefore, gender and ratings in individual assessment scales (Pain Catastrophizing Scale, Interpersonal Reactivity Index, Pain Anxiety Symptoms Scale, Behavioral Approach/Inhibition System) were used as covariates. To account for violations of sphericity, the Greenhouse-Geisser procedure was used to correct degrees of freedom.

## Results of Experiment 2

The structure of the results of the middle part of experiment 2 (main experiment of factor *Block*) was similar to that of experiment 1. ANCOVA for valence ratings revealed significant main effects for *Prime* ($F_{(1; 22)} = 12.56$; p = 0.002; $\eta_p^2 = 0.36$), *Category* ($F_{(1.68; 36.9)} = 211.66$; p < 0.001; $\eta_p^2 = 0.91$), and *Word* ($F_{(5.2; 114.3)} = 8.80$; p < 0.001; $\eta_p^2 = 0.29$) as well as significant interactions for *Category*\**Word* ($F_{(9; 198.2)} = 5.76$; p < 0.001; $\eta_p^2 = 0.21$) and *Prime*\**Category* ($F_{(2.2; 47.5)} = 4.94$; p = 0.009; $\eta_p^2 = 0.11$). Interestingly, differing from experiment 1, there was no main effect for *Repetition* ($F_{(1.8; 40.1)} = 1.13$; p = 0.327; $\eta_p^2 = 0.05$). Regarding pain ratings, we found significant main effects for *Prime* ($F_{(1; 22)} = 174.63$; p < 0.001; $\eta_p^2 = 0.89$) and *Category* ($F_{(1.9; 41.2)} = 4.52$; p = 0.008; $\eta_p^2 = 0.10$), but no interaction. Post-hoc analyses regarding the significant interaction effects were similar to experiment 1 and because of that are not given in detail here.

The main reason for realizing this second experiment was to analyze changes in the valence ratings between the pre-block, the main experiment (just trials without painful stimulation), post-block 1, and post-block 2. ANCOVA revealed a significant main effect both for *Block* ($F_{(3; 66)} = 3.28$; p = 0.026; $\eta_p^2 = 0.13$) and *Category* ($F_{(3; 66)} = 205.92$; p < 0.001; $\eta_p^2 = 0.90$), but no interaction. The main effect for *Block* showed lower valence ratings pre-block vs. post-block 1 (t = -2.37; p = 0.027), main experiment vs. post-block 1 (t = -2.74; p = 0.012), and higher valence ratings post-block 1 vs. post-block 2 (t = 2.63; p = 0.015) (Fig 5). The other contrasts for the main effect for *Block*: pre-block vs. main experiment (t = -1.37; p = 0.190), pre-block vs. post-block 2 (t = -0.90; p = 0.380), and main experiment vs. post-block 2 (t = 0.03; p = 0.970). The main effect for *Category* showed only highly significant contrasts, except for the contrast pain-related words (mean valence rating (± S.E.) of 6.9 ± 0.14) vs. negative words (7.0 ± 0.14), which was not significant (t = -0.79; p = 0.437). Mean valence rating (± S.E.) for neutral words was 4.6 ± 0.11 and for positive words 2.4 ± 0.17 with following contrasts: higher valence ratings for pain-related words vs. neutral words (t = 12.91; p < 0.001), pain-related words vs. positive words (t = 16.17; p < 0.001), negative words vs. neutral words (t = 12.93; p < 0.001), negative words vs. positive words (t = 16.38; p < 0.001), and neutral words vs. positive words (t = 11.18; p < 0.001).

None of the various covariates (Gender, Pain Catastrophizing Scale, Interpersonal Reactivity Index, Pain Anxiety Symptoms Scale, Behavioral Approach/Inhibition System) had any significant effect on the results mentioned above.

## Discussion to Experiment 2

We were able to replicate most of the findings of experiment 1, especially those with respect to our primary aim and hypothesis. We will not replicate the discussion on these results here.

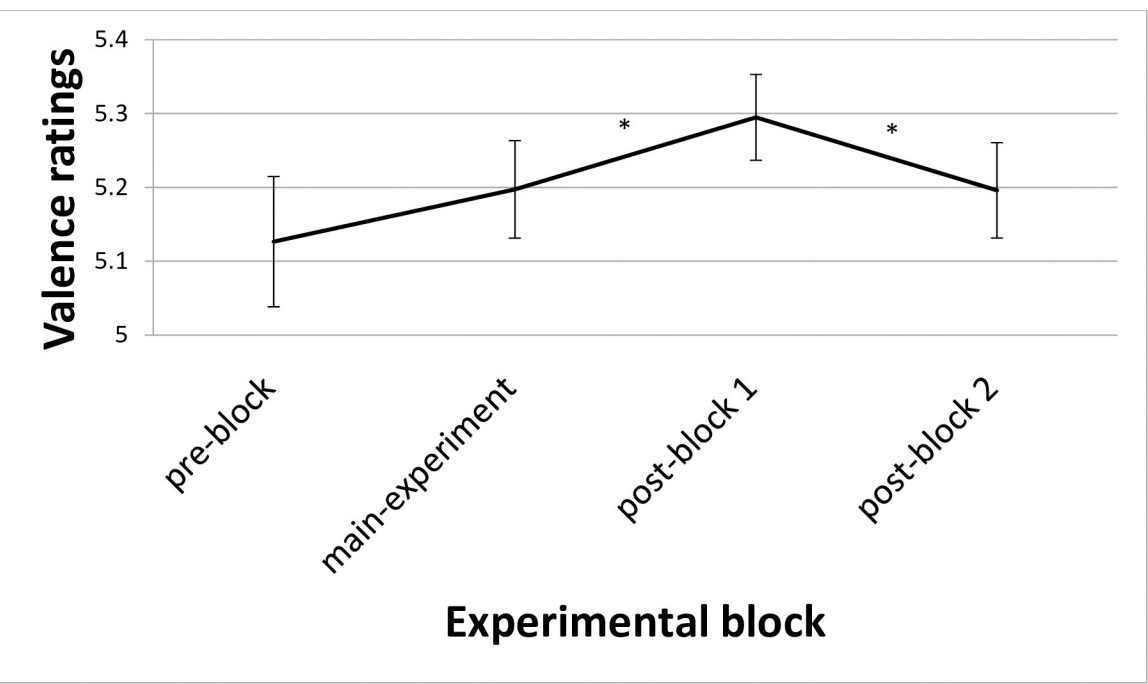

**Fig 5. Valence ratings with respect to the factor Block.** Data are reported as mean (SE); *: p < 0.05.

However, in contrast to experiment 1 we did not find any significant effects concerning factor *Repetition* in experiment 2. So, the question whether the repetition of a word influences valence ratings in combination with the application of a painful prime remains unanswered.

As one of the major goals of Experiment 2 was to assess the after-effects of the painful stimulation, we specifically selected and analyzed valence ratings for trials without painful stimulation (as the pre- and the post-blocks did not include trials with painful stimulation) and found a main effect of *Block*. The negative priming continued to affect the valence ratings even some minutes after the painful priming had stopped as shown in experiment 2. The overall valence ratings are most negative in post-block 1, after the painful electrical stimulation of the main experiment stopped (see Fig 5). So, the painful priming does not only affect valence ratings directly as short-term priming effect, but also leads to more negative valence ratings during a certain timeframe after the end of the priming. Long-term priming effects are known in several time scales so that this result is in line with previous research (for an overview on long-term priming, see [35]). After this time (i.e. in post-block 1), valence ratings decreased almost to baseline level in post-block 2.

We found no significant effect of the various covariates on the results. The heeded covariates are known to influence pain ratings as mentioned above. On the one hand, we can exclude these variables as having influenced our results significantly. On the other hand, one might wonder why these variables influence pain ratings in other studies. The reason for this is probably that effects are quite low and are often visible only in larger groups [36,37]. Furthermore, we did not focus on pain ratings in this study, but on valence ratings to target words. This might be another reason for the absence of effects of the covariates.

## Overall discussion

We investigated the effect of painful stimuli on valence ratings of succeedingly presented adjectives of different meaning and observed a priming effect of painful stimuli on the valence

of adjectives. Importantly, this effect is more pronounced for pain-related words compared to negative, pain-unrelated words. The priming effect continued for a several minutes after the painful priming had stopped.

Both experiments reveal a priming effect of painful stimuli on valence ratings of succeedingly presented adjectives irrespective from the meaning of the word (Fig 2). This effect is in line with the emotional priming theory [14] as painful stimuli usually induce negative emotions [38]. This change in valence was visible for all categories of adjectives (significant for neutrals when concatenating both experiments).

The most important result with respect to the primary aim of this study is the more pronounced priming effect of painful stimuli on valence ratings to succeedingly presented pain-related adjectives as compared to negative adjectives. This was found in both experiments. As mentioned earlier, painful primes do not only lead to a simple emotional priming effect similar for negative and pain-related adjectives as predicted from emotional priming theory [14], but to an additional priming effect of noxious stimulation on valence ratings to pain-related adjectives. This additional priming effect could be explained by the theory of neural networks [16]. An associative memory network will be developed as a result of past pain experiences. This network would strengthen its connections and increase its efficacy whenever we are exposed to real or potentially real painful stimuli, or also to stimuli that semantically represent harm or threat [12]. Thus, painful stimuli used in our experiment as primes could activate specific pain-related neural networks. The additional activation of a semantic network associated specific pain-relatedness (pain-related adjectives) will probably lead to a stronger activation in this network compared to negative, but pain-unrelated primes. As negative valence is one component of pain processing, this will result in more negative valence ratings for pain-related adjectives. In a similar direction, Becker et al. [39] also suggested a neural network model to explain long-term priming effects. They proposed that each time a word is processed on a semantic level, this processing will result in light increase of weights to assess this network. Combined with the priming by physical stimuli that more or less specifically activate such a network (at least, more than just networks for negative adjectives), this stronger or deeper activation might result in a more negative valence. Moreover, our results are in line with studies investigating the effect of semantic primes on painful stimuli [e.g. 9–11,17,18,40]. In these studies, primes with negative valence (pain-related and negative) not only increased pain ratings for following pain stimuli, but also pain stimuli increase the negative valence of following semantic pain-related and negative stimuli. Therefore, this could describe a vicious circle for the origin, development and perpetuation of chronic pain. Felt pain might be described by patients with pain-related words, which increases internal pain ratings. In turn, these increased pain ratings might lead to more negative internal valence ratings of pain-related words, which again might lead to an increase of internal pain ratings [41]. Of course, more detailed research is required to support this hypothesis, especially in patients [40,41]. While positive words were rated less positive after painful stimulation in accordance with the motivational priming theory [14], it was one unexpected result of this study that the negative priming effect of painful stimulation was maximal for positive words in both experiments (Fig 3). Several reasons might account for this result. On the one hand, positive words are probably most unexpected with respect to their valence. From the point of view of theory of predictive coding [42–44], there is less prediction error with respect to valence after a painful stimulus (evoking negative valence) in a plausible order from pain-related and negative over neutral to positive adjectives. In contrast, this prediction error is different in trials without a painful stimulus. Here, the discrepancy is minimal for neutral adjectives and roughly similar for the other types of adjectives as they all have the same valence difference to neutral adjectives. One might discuss whether in these trials the prediction is somewhat between neutral and negative as only trials occur

without or with painful stimulation before presentation of an adjective. In any case, a prediction error will lead to an activation of higher centers in processing hierarchy, requesting for salience and to actualization of expectations. This results in larger cognitive load which might have effects on valence (and the perception of pain). On the other hand, the age of acquisition (AoA) might have influenced our results. AoA is an important variable because there exists a negative correlation between AoA and valence, i.e., positive words tend to be acquired earlier in life than negative words [45–48]. This effect would possibly also allow to provide a deeper network for positive words which per se might lead to deeper processing. This deeper processing together with the discrepancy of valence described above might also explain the unexpected priming result of painful stimulation to positive adjectives. Unfortunately, no German database containing the AoA of our adjectives is available. So, to test this possible effect, we conducted a rating study on a sample of 32 native German speaking university students and postgraduates (18 female and 14 male, 26.6 ± 4.3 years old). Participants were asked to rate the AoA of all words used in this study. Six versions of the questionnaire with different randomizations were administered to ensure the absence of order effects. Instructions and questionnaire were similar to the study of Birchenough, Davies, and Connelly [49]. With respect to AoA, we found no significant effect of *Category* ($F_{(3; 93)} = 1.57$; $p = 0.202$; $\eta_p^2 = 0.05$). So, the AoA of the adjectives used in this study do not seem to differ significantly with respect to *Category* (AoA mean (± S.E.) of 5.6 ± 0.10 for pain-related words, 5.3 ± 0.11 for negative words, 5.5 ± 0.09 for neutral words, and 5.4 ± 0.08 for positive words).

## Limitation and future directions

The current study shows clear systematic priming effects; however, effect sizes are relatively small. This may be because, in the particular group of healthy young individuals, the modulation of the perception of painful stimuli is less influenced by painful primes. Studies indicate a more pronounced priming effect of pain-related primes versus negative, pain-unrelated primes in patients with chronic pain compared to healthy controls [1,17,40]. It appears that associative learning networks regarding pain experiences are stronger in patients with chronic pain compared to healthy controls. Therefore, chronic pain patients show larger priming effects for pain-related primes versus negative, pain-unrelated primes. Future studies might investigate the effects in patients with chronic pain.

Only electrical painful stimuli were used as primes in the present study. The application of a greater variety of painful primes (e.g. heat, cold, pressure) could generalize the findings for other qualities of pain.

The sample of the current study consists only of healthy young people. As mentioned above, further investigations should focus on chronic pain patients.

In experiment 1, we found higher valence ratings in repetition 4 and repetition 5 in contrast to repetition 1. In contrast, we did not find significant effects concerning factor *Repetition* in experiment 2. Future research should investigate a possible order effect and clarify the question whether it was a random effect in experiment 1 or not.

Originally, the adjectives used in this study were matched with respect to word frequency, length, number of syllables, absolute amount of valence, arousal, and unambiguousness [18]. AoA for the adjectives were measured in a separate sample with different participants. In future studies, AoA as well as other variables like abstractness or imageability should be taken into account when selecting the adjectives used as stimuli because these other parameters might have influence on the results, too.

We used various questionnaires known to influence pain processing as moderating variables and found no influence. As mentioned above, this might be due to the fact that these

variables influence pain ratings, not valence ratings. In future studies, other more suitable variables could be identified and considered. Furthermore, analysis of the recall test after the experiment revealed that pain-related words were recalled significantly less compared to negative, neutral, or positive words. This might hint a selective memory effect for each of the word categories, e.g. by AoA (as stated earlier). Future research could investigate the existence and character of such an effect.

## Conclusion

In summary, the motivational priming theory and the theory of neural networks are also applicable to painful primes combined with semantic stimuli. The current study showed the influence of painful primes on perception and processing of the valence of words. Painful primes lead to more negative valence ratings of pain-related and negative, pain-unrelated words in contrast to neutral words. This effect is stronger for pain-related compared to negative, pain-unrelated words and lasts even for some minutes after the painful priming stopped.

## Supporting information

**S1 Table. Valence ratings (mean ± SD) of each word in the main experiment.**
(PDF)

**S2 Table. Corrected p-values for contrasts regarding interaction effect Category$^*$Word in the main experiment.**
(PDF)

## Acknowledgments

We thank Mr. Holger Hecht for his valuable work in the technical preparation of this study, MSc Maria Geisler and BSc Ani Tamir Abou Seif for their help during the preparation of the revision of the manuscript.

## Author Contributions

**Conceptualization:** Christoph Brodhun, Thomas Weiss.

**Data curation:** Christoph Brodhun, Thomas Weiss.

**Formal analysis:** Christoph Brodhun, Thomas Weiss.

**Investigation:** Christoph Brodhun, Eleonora Borelli.

**Methodology:** Christoph Brodhun, Eleonora Borelli, Thomas Weiss.

**Project administration:** Christoph Brodhun, Thomas Weiss.

**Supervision:** Thomas Weiss.

**Writing – original draft:** Christoph Brodhun.

**Writing – review & editing:** Christoph Brodhun, Eleonora Borelli, Thomas Weiss.

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
