## [Decision Letter · Decision Letter 0]

20 Oct 2020

PONE-D-20-29181

Influence of acute pain on valence rating of words

PLOS ONE

Dear Dr. Weiss,

Thank you for submitting your manuscript to PLOS ONE. Two expert reviewers have now provided comments about your manuscript. While both reviewers had positive words about your study, they also raised several issues that i would like that you consider in a revised version of your work. After my own reading of the manuscript, i would like that you pay particular attention to the lack of description of the motivational priming theory and the possible age of acquisition confound. Also, i encourage the authors to share the stimulus materials used in their study.

We look forward to receiving your revised manuscript.

Kind regards,

José A Hinojosa, Ph.D.

Academic Editor

PLOS ONE

Journal Requirements:

2. Thank you for including your ethics statement:  "We received approval from the Ethics Committee of the Faculty of Social and Behavioural Sciences of the Friedrich Schiller University Jena; Vote No. FSV 14/04.".   

a. Please provide additional details regarding participant consent. In the ethics statement in the Methods and online submission information, please ensure that you have specified what type you obtained (for instance, written or verbal, and if verbal, how it was documented and witnessed). If your study included minors, state whether you obtained consent from parents or guardians. If the need for consent was waived by the ethics committee, please include this information.

Reviewers' comments:

Reviewer's Responses to Questions

**Comments to the Author**

1. Is the manuscript technically sound, and do the data support the conclusions?

Reviewer #1: Partly

Reviewer #2: Yes

2. Has the statistical analysis been performed appropriately and rigorously? 

Reviewer #1: No

Reviewer #2: Yes

3. Have the authors made all data underlying the findings in their manuscript fully available?

Reviewer #1: No

Reviewer #2: No

4. Is the manuscript presented in an intelligible fashion and written in standard English?

Reviewer #1: Yes

Reviewer #2: Yes

5. Review Comments to the Author

Reviewer #1: Review of the manuscript PONE-D-20-29181titled ‘Influence of acute pain on valence rating of words’

The present work analyzes how painful primes can influence the valence of semantic stimuli. In two experiments, participants were asked to rate the valence of different semantic stimuli (pain-related, negative, positive, and neutral adjectives) after they were primed with aversive electrical stimuli. The results showed that valence ratings of pain-related, negative, and positive words were more negative after the electrical prime. This priming effect continued even some minutes after the prime ends.

1. General comments and overall evaluation

Although the study is in general well done, it needs to be improved before publication. First, the general structure with combining sections of experiment 1 and 2, and not presenting them in separate blocks, is sometimes confusing and makes the main plot of the paper makes hard to follow. More importantly, there are some critical methodological and theoretical aspects in which the current version of the paper requires further consideration and need to be revised.

On the other hand, I think the paper is well written although I am not native speaker of English.

2. Detailed review of the different sections of the paper

Introduction

Lines 39-40. I encourage to the authors to extend the description of the motivational priming theory, which is now described only in a sentence.

The effects described in the lines 41-43 would be better understood with some examples of prime-target pairs.

Material and methods

Subjects

I only suggest the use of the term “participants” to title this section. The participant description is well done.

Pain stimuli

Pain stimuli and its calibration are clearly described.

Study design

This section is a mix of “word stimuli” and “procedure”, which I think can be confusing for the reader, so I would recommend to make separate sections for materials (words and pain stimuli), instruments (scales and current stimulator descriptions), and for procedure (task events and instructions to participants), by each experiment.

The word set is not presented as appendix or in any separate file. I understand that the authors will present the stimuli if the paper is accepted for publication as the they have committed to make all data available.

Regarding the controls on the used materials (words), I strongly miss the AoA (age of acquisition) as a matching variable. There is since some years enough evidence to consider AoA of words as important as other variables like word frequency or length in word recognition and naming tasks, among others, and critically, a statistically significant, negative correlation between AoA and valence (i.e., positive words tend to be early-acquired in life) has been consistently reported in many norms studies (e.g., Imbir, 2016; Stadthagen-Gonzalez, 2017; Warriner et al., 2013). Note that the correlation is moderate (around -.2), but it is overall higher than the correlation between lexical frequency and valence. Therefore, AoA should empirically be more relevant than frequency in the word selection of studies about emotional valence. So, I am afraid that the authors should discard a confound effect because of the possible differences in AoA between word categories in both experiments. They can collect AoA data from previous databases (Birchenough, Davies, & Connelly, 2017) or collect new AoA ratings if needed, and then compare the mean AoA of words across word categories (pain-related, negative, etc.).

Each word was presented ten times to each participant in exp.1 and I understand that some times more in exp.2, so an order effect is expected in each experiment. The words preceded by pain primes were likely negatively conditioned and therefore they can be more negative rated in subsequent trials. This potential order effect should be discarded by including trial repetition as a factor (or covariable) in the analysis.

Measures in Experiment 2.

Here one run into a number of measures by questionnaires that are not mentioned until this section. I think that the rationale of these measures in relation with the objectives of the study should be commented in the introduction.

Data analyses

I think this section should go in results. Apart of that, I think that some descriptive statistics or plots (histograms, density plots, etc.) about the dependent variables’ distribution should be reported in each experiment in order to see if there is a normal distribution. This should also be done for the scores obtained in scales used in the exp.2 as. Alternatively (or additionally), the authors could perform normality tests in the results section.

Results

All performed analyses are by participants and, having into account that there few items (10) in each category of words, I strongly recommend to carry out an analysis by items in order to know if the effects are generalizable to the rest of words with similar characteristics.

Experiment 1

All t values (and df) in each post-hoc test, statistically significant or not, should be reported and, when necessary, Bonferroni corrections should be applied.

The ANOVA on the number of recalled words by word category shows differences between some categories, but this does not mean a direct relation with the priming effect. I mean that the differences can be attributed to the word characteristics themselves as other uncontrolled but relevant variables (like AoA or imageability) might be affecting the performance of this task. This is mentioned in the limitations section with a succinct sentence “This might hint a selective memory effect” and maybe it can be expanded.

Experiment 2

Same comments about t values and Bonferroni corrections as in the experiment 1.

ANOVA should be changed by ANCOVA (line 259)

Additionally, there are some considerations about the covariates. First, “sex” was not announced to be a covariate until here, at line 272. If the authors want to control the sex (or gender) of participants, first they have to explain why, and second I consider that it is more correct to including this variable as a dummy factor to adjust the degrees of freedom of the rest of contrasts than treat it as a covariate (although this is not formally incorrect). On the other hand, the non-significant effects of any covariate seem to be surprising because it was supposed they were moderating valence ratings. Moreover, I wonder if the covariates actually make a contribution in the model. In other words, ¿would do we get the same results without the covariates? If the ANOVA (without covariates) shows different results of those from the performed ANCOVA, it is possible to conclude that the covariates actually matter because they have an effect, but this was not strong enough to be detected in this experiment. All this should be reviewed and discussed by the authors

The t values of the post hoc comparison for Block (lines 262-263) should be revised because all of them are positive and I think that there should be positive and negative values if the order of contrasted levels of Block indicate the mean differences.

Discussion

In general, I expected a more theoretical discussion about implications of the effects.

Experiment 1

It should be interesting to discuss in a deeper way the implications of the results under the motivational priming and the neural networks theories and, if possible, with a specification of the cognitive mechanisms involved in the priming effects found.

The explanation that the incongruency generated by the pairs painful stimulation-positive words might be the cause of the higher priming effect on valence ratings of this type of words compared to the other categories is speculative (the authors only cite a study about the N400 that has a very indirect relationship with the present study) and partial. The authors should explain why neutral words do not generate an incongruency when primed by painful stimulation. I do not mean that neutral words should generate an incongruency in the same magnitude than positive words, but why authors assume that neutral words do not generate incongruency at all.

Experiment 2

The sentence regarding that only trials without painful stimulation were used in the analyses (lines 311-312) would go better in results than here

Text in parenthesis at line 314 should go in procedure.

The authors re-describe the results but not explain why negative priming affected valence ratings for some minutes after the end of priming.

Limitations

The paragraph from lines 345 to 355 fits better in the discussion than here.

Typing mistakes:

Line 130, “á” >> of

Used references

Birchenough, J. M., Davies, R., & Connelly, V. (2017). Rated age-of-acquisition norms for over 3,200 German words. Behavior research methods, 49(2), 484-501.

Imbir, K. K. (2016). Affective Norms for 4900 Polish Words Reload (ANPW_R): Assessments for Valence, Arousal, Dominance, Origin, Significance, Concreteness, Imageability and, Age of Acquisition. Frontiers in Psychology, 7. https://doi.org/10.3389/fpsyg.2016.01081

Stadthagen-Gonzalez, H., Imbault, C., Pérez Sánchez, M. A., & Brysbaert, M. (2017). Norms of valence and arousal for 14,031 Spanish words. Behavior Research Methods, 49(1), 111-123. https://doi.org/10.3758/s13428-015-0700-2

Warriner, A. B., Kuperman, V., & Brysbaert, M. (2013). Norms of valence, arousal, and dominance for 13,915 English lemmas. Behavior Research Methods, 45(4), 1191-1207. https://doi.org/10.3758/s13428-012-0314-x

Reviewer #2: The present manuscript of Brodhum et al. investigates the influence of painful primes on the perception of the valence of subsequent semantic stimuli by two experiments. The first one, explored the existence of the effect of this influence, and second one, focused on the length of the effect. The behavioral data indicate more negative valence ratings of the emotional words (pain-related, negative and positive) after painful prime in contrast to no prime. In addition, the effect continued some minutes after the painful prime. This is a potentially interesting study since it addresses a topic that has not been studied enough up to date. Thus, while the overall research question is within the scope of PLOS ONE readers, some aspects should be need to be improved.

-The introduction section is clear and concise. However, despite using positive words in the experiments, are not information about this kind of stimuli, and how the motivational priming theory explains its possible effects. Actually, this theory indicated two opposite responses for positive versus negative stimuli. This information its important, not only to explain the use of positive words in this study. Also, because of the subsequent results that to some extent contradict the theory.

In addition, since the authors want to explore the duration of the effect, should be include previous information about this point. Or, if there are not previous evidence, indicate it.

On the other hand, since several tests were applied to evaluate different variables that could be mediating the results, it is necessary to indicate in this section how these variables can modify the effect studied.

- In the Method section, authors did not mention any method used for the calculation of the sample size. Where there a priori power analyses to support adequate power for statistical analyses?

Additionally, since working with words it is useful to know the educational level of the subjects.

Please include examples of the different selective adjectives. Positive words were balance in arousal to negative words? Please give some explanation about this aspect.

In the experiment 2, please, include the scale used to measure the valence. There are not information about the duration of the each block and the duration of the length of the interval between the blocks.

Please include the duration of the recording session and where were conducted.

Questionnaires: authors did not indicate why is important or the relation between all the tests and the variables studied. In addition, should be include a more developed explanation of the constitution of each scale used.

-In the Discussion section:

The authors indicate that the results in experiment 1 are in line with the motivational priming theory, however, as I indicated earlier, this theory considers opposite effects for negative and positive stimuli. If the negative prime (painful stimulation) has an effect in the valence of the words, why are not influence in neutral words but if the positive?

In addition, not all results are discuses, like why positive words were evaluated so much more negatively than negative ones? The motivational priming hypothesis indicates that in a negative context, the negative is exacerbated. In line with this, negative word under painful condition should be evaluate as more negative. However, the opposite effect was observed, where negative words are evaluated less negatively.

Furthermore, suddenly, a study with PER is briefly described to try to explain the effect observed in the positive words. However, the explanation is unclear and departs from the methodology used. Should we have an N400 for this task? What process would be reflecting here?

At the end of the first study, the authors indicated that not finding differences between words related to pain and negative ones was expected. However, this contradicts what was previously stated in the introduction section, lines 58 to 60.

In the case of experiment 2, there are not a discussion, just an explanation of the results. This part should be improved.

6. PLOS authors have the option to publish the peer review history of their article (what does this mean?). If published, this will include your full peer review and any attached files.

Reviewer #1: **Yes: **Miguel Á. Pérez-Sánchez

Reviewer #2: No

---

## [Author Response · Author response to Decision Letter 0]

28 Dec 2020

Responses to Journal Requirements and Reviewers 

We thank the reviewers for taking time and investing effort to provide their constructive comments to improve the manuscript (Ms). We have addressed all the comments in a point-by-point reply. Our answers are indicated in blue font below. We provide a marked-up copy of our manuscript highlighting changes made in the original version as well as an unmarked version of the revised manuscript. We report places of changes in this letter with respect to the marked-up copy of Ms. 

Journal Requirements

Answer: We followed this advice. 

2. Thank you for including your ethics statement: "We received approval from the Ethics Committee of the Faculty of Social and Behavioural Sciences of the Friedrich Schiller University Jena; Vote No. FSV 14/04.". 

a. Please provide additional details regarding participant consent. In the ethics statement in the Methods and online submission information, please ensure that you have specified what type you obtained (for instance, written or verbal, and if verbal, how it was documented and witnessed). If your study included minors, state whether you obtained consent from parents or guardians. If the need for consent was waived by the ethics committee, please include this information.

Answer: We thank for this advice. We give now additional information in the ethical statement and the MS, as requested.

 Answer: We will make available all data at acceptance. Please note that we provide already substantial part of the data in the supplementary material. 

Reviewer 1

The present work analyzes how painful primes can influence the valence of semantic stimuli. In two experiments, participants were asked to rate the valence of different semantic stimuli (pain-related, negative, positive, and neutral adjectives) after they were primed with aversive electrical stimuli. The results showed that valence ratings of pain-related, negative, and positive words were more negative after the electrical prime. This priming effect continued even some minutes after the prime ends.

1. General comments and overall evaluation

Although the study is in general well done, it needs to be improved before publication. First, the general structure with combining sections of experiment 1 and 2, and not presenting them in separate blocks, is sometimes confusing and makes the main plot of the paper makes hard to follow. …

Answer: We thank for this advice. In accordance with your suggestion, we changed the order by binding into separate blocks each of the two experiments. So, we hope that the Ms is easier to follow now. 

… More importantly, there are some critical methodological and theoretical aspects in which the current version of the paper requires further consideration and need to be revised.

On the other hand, I think the paper is well written although I am not native speaker of English.

The details of the criticism of the reviewer are explained in the second part of review so that we will answer to each point in part two.

2. Detailed review of the different sections of the paper

Introduction

Lines 39-40. I encourage to the authors to extend the description of the motivational priming theory, which is now described only in a sentence.

Answer: We added some more information concerning motivational priming theory (in accordance with the reviewer’s suggestion, now lines 39-46). 

The effects described in the lines 41-43 would be better understood with some examples of prime-target pairs.

Answer: We thank for this advice. In accordance with your suggestion, we included examples of the primes (now lines 49-54).

Material and methods

Subjects

I only suggest the use of the term “participants” to title this section. The participant description is well done.

Answer: We followed this advice (line 87). 

Pain stimuli

Pain stimuli and its calibration are clearly described.

Thank you.

Study design

This section is a mix of “word stimuli” and “procedure”, which I think can be confusing for the reader, so I would recommend to make separate sections for materials (words and pain stimuli), instruments (scales and current stimulator descriptions), and for procedure (task events and instructions to participants), by each experiment.

Answer: We reorganized the structure with respect to several aspects: First, we organized the middle of the Ms (Methods, Results) according to suggestions at general comments, i.e. first Exp. 1, second Exp. 2. Within the description of the experiments, we tried to follow the proposal made here, i.e. with separate sections for Word stimuli and Study design. 

The word set is not presented as appendix or in any separate file. I understand that the authors will present the stimuli if the paper is accepted for publication as the they have committed to make all data available.

Answer: As we understand, the reviewer requests the stimulus material (instead of pointing to this information as in the references). We now gave examples of words in the text and show the stimulus material (word set) in the supporting information together with the major results (Supplementary table S1: Valence ratings (mean ± SD) of each word in the main experiment.). 

Regarding the controls on the used materials (words), I strongly miss the AoA (age of acquisition) as a matching variable. There is since some years enough evidence to consider AoA of words as important as other variables like word frequency or length in word recognition and naming tasks, among others, and critically, a statistically significant, negative correlation between AoA and valence (i.e., positive words tend to be early-acquired in life) has been consistently reported in many norms studies (e.g., Imbir, 2016; Stadthagen-Gonzalez, 2017; Warriner et al., 2013). Note that the correlation is moderate (around -.2), but it is overall higher than the correlation between lexical frequency and valence. Therefore, AoA should empirically be more relevant than frequency in the word selection of studies about emotional valence. So, I am afraid that the authors should discard a confound effect because of the possible differences in AoA between word categories in both experiments. They can collect AoA data from previous databases (Birchenough, Davies, & Connelly, 2017) or collect new AoA ratings if needed, and then compare the mean AoA of words across word categories (pain-related, negative, etc.).

Answer: Thank you for this advice. To be honest, we were not aware of this factor. Unfortunately, the German database does not include our adjectives, so that we were not able to compare this aspect for our stimulus material. We believe that an additional examination for AoA would overburden us and overcharge the Ms. Moreover, as we understand, the differences are mainly between positively vs. negatively valanced words. Therefore, AoA is quite plausible for explaining differences in priming effects between positive and negative valanced words. We included this important point to the discussion of the effect of positive adjectives (lines 650-657). Moreover, we included it as an unexplained factor to the limitation section (687-703). 

Each word was presented ten times to each participant in exp.1 and I understand that some times more in exp.2, so an order effect is expected in each experiment. The words preceded by pain primes were likely negatively conditioned and therefore they can be more negative rated in subsequent trials. This potential order effect should be discarded by including trial repetition as a factor (or covariable) in the analysis.

Answer: Thank you very much for mentioning this important point. While we do not think that conditioning plays an important role (as each of the words is presented in 50% of trials with and in 50% of trials without preceding pain stimulation), repetitions might play a role. Therefore, we now included an additional within-subject factor Repetition to the ANOVA. While Repetition is relevant for the data, the main results with respect to our theoretical framework are not changed by this additional factor. We now show the results including factor Repetition (lines 255ff). 

Measures in Experiment 2.

Here one run into a number of measures by questionnaires that are not mentioned until this section. I think that the rationale of these measures in relation with the objectives of the study should be commented in the introduction.

Answer: We apologize for not making the sense of including questionnaires to Exp. 2 clear. 

The questionnaires were included as the different characteristics examined with the questionnaires are known to possibly influence on pain ratings. After having found the principle correctness of our primary hypotheses in Exp. 1, the primary aim of including these questionnaires to Exp. 2 was to control for possible influences on our results. We explained this now in more detail in the description of experiment 2 (lines 418ff). 

Data analyses

I think this section should go in results. Apart of that, I think that some descriptive statistics or plots (histograms, density plots, etc.) about the dependent variables’ distribution should be reported in each experiment in order to see if there is a normal distribution. This should also be done for the scores obtained in scales used in the exp.2 as. Alternatively (or additionally), the authors could perform normality tests in the results section.

Answer: According to the recommendations to reorganize the structure, we moved the section data analysis to each experiment. We also included the mentioning of tests for normality of data (before ANOVA or ANCOVA). This is now included to the Ms (lines 245ff). However, we believe that the description of the statistical analyses belongs to the method section. We organized it in such a way that it directly precedes the result section in order to not repeat this content several times. 

Results

All performed analyses are by participants and, having into account that there few items (10) in each category of words, I strongly recommend to carry out an analysis by items in order to know if the effects are generalizable to the rest of words with similar characteristics.

Answer: We followed the advice and included an additional factor (Repetition) to the analysis. We also provide a supplementary table with each item including its characteristics. However, we remained with Word categories as this was our major research interest. Moreover, our main research questions and hypotheses belong to the Word categories. 

Experiment 1

All t values (and df) in each post-hoc test, statistically significant or not, should be reported and, when necessary, Bonferroni corrections should be applied.

Answer: According to advice, we included now all statistical values (throughout the MS). 

The ANOVA on the number of recalled words by word category shows differences between some categories, but this does not mean a direct relation with the priming effect. I mean that the differences can be attributed to the word characteristics themselves as other uncontrolled but relevant variables (like AoA or imageability) might be affecting the performance of this task. This is mentioned in the limitations section with a succinct sentence “This might hint a selective memory effect” and maybe it can be expanded.

Answer: As described earlier, we included now AoA to the discussion of priming effects on positive adjectives. Additionally, we included this possible effect as requested and possible other effects (abstractness, imageability) to the limitation section (lines 690ff). We also added an expansion of the selective memory effect as requested (lines 701f).

Experiment 2

Same comments about t values and Bonferroni corrections as in the experiment 1.

Answer: According to advice, we included now all statistical values (throughout the ms).

ANOVA should be changed by ANCOVA (line 259)

Answer: We changed this accordingly.

Additionally, there are some considerations about the covariates. First, “sex” was not announced to be a covariate until here, at line 272. If the authors want to control the sex (or gender) of participants, first they have to explain why, and second I consider that it is more correct to including this variable as a dummy factor to adjust the degrees of freedom of the rest of contrasts than treat it as a covariate (although this is not formally incorrect). On the other hand, the non-significant effects of any covariate seem to be surprising because it was supposed they were moderating valence ratings. Moreover, I wonder if the covariates actually make a contribution in the model. In other words, ¿would do we get the same results without the covariates? If the ANOVA (without covariates) shows different results of those from the performed ANCOVA, it is possible to conclude that the covariates actually matter because they have an effect, but this was not strong enough to be detected in this experiment. All this should be reviewed and discussed by the authors

Answer: We apologize for this mistake. Gender was included as it is known to possibly influence on pain ratings. We explained this in detail (lines 418ff, 547ff).

The t values of the post hoc comparison for Block (lines 262-263) should be revised because all of them are positive and I think that there should be positive and negative values if the order of contrasted levels of Block indicate the mean differences.

Answer: We are sorry, we reported the absolute values of the t-values. We corrected this mistake (throughout the MS).

Discussion

In general, I expected a more theoretical discussion about implications of the effects.

Answer: We added more discussion on implications now (e.g., lines 588ff). Details are specified below. 

Experiment 1

It should be interesting to discuss in a deeper way the implications of the results under the motivational priming and the neural networks theories and, if possible, with a specification of the cognitive mechanisms involved in the priming effects found.

Answer: We provide now a deeper discussion on both theories. We tried to realize a specification with respect to some cognitive mechanisms (lines 593ff). In case, this reviewer has different aspects in mind, we will be happy to include these after receiving a little help for this.

The explanation that the incongruency generated by the pairs painful stimulation-positive words might be the cause of the higher priming effect on valence ratings of this type of words compared to the other categories is speculative (the authors only cite a study about the N400 that has a very indirect relationship with the present study) and partial. The authors should explain why neutral words do not generate an incongruency when primed by painful stimulation. I do not mean that neutral words should generate an incongruency in the same magnitude than positive words, but why authors assume that neutral words do not generate incongruency at all.

Answer: We agree. We tried to integrate this point with respect to the theory on predictive coding (lines 662-673). From this point of view, there is less prediction error with respect to evoked valence for a following painful stimulus in a plausible order from pain-related to negative to neutral to positive adjectives. In contrast, this prediction error is different for the trials without a following painful stimulus. Here, the discrepancy is minimal for neutral adjectives and similar for the other types of adjectives as they all have the same valence. In any case, a prediction error will lead to an activation of salience and to actualization of expectations. This results in larger cognitive load which might have effects on the valence ratings and the perception of pain. As a result (but highly speculative and not included to the MS), there might be counteracting activity for network priming vs. prediction errors. We included this into the general discussion section (lines 639-657). 

Experiment 2

The sentence regarding that only trials without painful stimulation were used in the analyses (lines 311-312) would go better in results than here

Answer: This sentence was misleading. On the one hand, we analyzed the both trials with and without painful stimulation in Experiment 2. On the other hand, we assessed aftereffects of the painful stimulation (i.e. one of the major goals of Exp. 2), we analyzed specifically all trials without a following painful stimulation. It turns out that the overall valence ratings are more negative in the post-block 1 compared to baseline, while returning to baseline in post-block 2. Obviously, we need the sentence to clarify the selection of trials for this specific analysis at this point, so we cannot remove it from this part of text. However, we tried to make this point of selection of this specific analysis clearer now (lines 527-542).

Text in parenthesis at line 314 should go in procedure.

Answer: We moved this text to procedure as suggested (now end of §2 in the description of Exp. 2).

The authors re-describe the results but not explain why negative priming affected valence ratings for some minutes after the end of priming.

Answer: We formulated our explanation with respect to long-term priming in more detail (now lines 559ff).

Limitations

The paragraph from lines 345 to 355 fits better in the discussion than here.

Answer: We fully agree, thanks for this advice. We move this part to the discussion section.

Typing mistakes:

Line 130, “á” >> of

Answer: Thanks for careful reading. We changed it.

Used references

Birchenough, J. M., Davies, R., & Connelly, V. (2017). Rated age-of-acquisition norms for over 3,200 German words. Behavior research methods, 49(2), 484-501.

Imbir, K. K. (2016). Affective Norms for 4900 Polish Words Reload (ANPW_R): Assessments for Valence, Arousal, Dominance, Origin, Significance, Concreteness, Imageability and, Age of Acquisition. Frontiers in Psychology, 7. https://doi.org/10.3389/fpsyg.2016.01081

Stadthagen-Gonzalez, H., Imbault, C., Pérez Sánchez, M. A., & Brysbaert, M. (2017). Norms of valence and arousal for 14,031 Spanish words. Behavior Research Methods, 49(1), 111-123. https://doi.org/10.3758/s13428-015-0700-2

Warriner, A. B., Kuperman, V., & Brysbaert, M. (2013). Norms of valence, arousal, and dominance for 13,915 English lemmas. Behavior Research Methods, 45(4), 1191-1207. https://doi.org/10.3758/s13428-012-0314-xI have the following concerns: 

Answer: Thank you. We included those references to the discussion.  

Reviewer 2

The present manuscript of Brodhum et al. investigates the influence of painful primes on the perception of the valence of subsequent semantic stimuli by two experiments. The first one, explored the existence of the effect of this influence, and second one, focused on the length of the effect. The behavioral data indicate more negative valence ratings of the emotional words (pain-related, negative and positive) after painful prime in contrast to no prime. In addition, the effect continued some minutes after the painful prime. This is a potentially interesting study since it addresses a topic that has not been studied enough up to date. Thus, while the overall research question is within the scope of PLOS ONE readers, ….

Answer: Thank you for this kind assessment.

… some aspects should be need to be improved.

-The introduction section is clear and concise. However, despite using positive words in the experiments, are not information about this kind of stimuli, and how the motivational priming theory explains its possible effects. Actually, this theory indicated two opposite responses for positive versus negative stimuli. This information its important, not only to explain the use of positive words in this study. Also, because of the subsequent results that to some extent contradict the theory.

Answer: We now included a more extensive description of the motivational priming theory as well as predictions from this theory with respect to our categories of adjectives (lines 39-54).

In addition, since the authors want to explore the duration of the effect, should be include previous information about this point. Or, if there are not previous evidence, indicate it.

Answer: We also added information on the duration that we were able to identify (line 82f). 

On the other hand, since several tests were applied to evaluate different variables that could be mediating the results, it is necessary to indicate in this section how these variables can modify the effect studied.

Answer: We describe the variables now in the participants section as these variables are rather known to influence pain perception. We thought that the description is clearer at this place (now lines **) than in the intro, as it is rather to control other influences than to develop the theoretical point of view and/or aims and hypotheses for this study.

- In the Method section, authors did not mention any method used for the calculation of the sample size. Where there a priori power analyses to support adequate power for statistical analyses?

Answer: We used a tool for our a priori analyses (Hemmerich, W. (2020). StatistikGuru: Stichprobengröße für die ANOVA mit Messwiederholung berechnen. Retrieved from https://statistikguru.de/rechner/stichprobengroesse-anova-mit-messwiederholung.html). Considering the fact that there are no studies investigating the effect of painful priming on the valence of words, we used previous results of our research group investigating the effect of affective priming on pain ratings as a starting point. Considering the number of measurements for each person we calculated a necessary sample size of approximately 20 participants with α = 0.05 and power = 0.9 (lines 103-109).

Additionally, since working with words it is useful to know the educational level of the subjects.

Answer: All subjects were university students. We added this information in lines 84/85.

Please include examples of the different selective adjectives. Positive words were balance in arousal to negative words? Please give some explanation about this aspect.

Answer: We included an example for each word to the text (lines 152f). Moreover, we added information to the verbal material and the type of matching (lines 154ff). Finally, all used adjectives (together with their results) as well as the original data are now in the Supporting information.

In the experiment 2, please, include the scale used to measure the valence. 

Answer: We used the same scales as in experiment 1 (Fig 1). We added this information in line 401.

There are not information about the duration of the each block and the duration of the length of the interval between the blocks.

Please include the duration of the recording session and where were conducted.

Answer: We included this information (Study design of Exp. 1 and 2, first §).

Questionnaires: authors did not indicate why is important or the relation between all the tests and the variables studied. In addition, should be include a more developed explanation of the constitution of each scale used.

Answer: We included more information on the Questionnaires. The major reason to use them was to exclude possible influences on the results. Each of the examined variables has previously shown to possibly influence on pain ratings. 

We now included more information to make this point clear (418-427).

-In the Discussion section:

The authors indicate that the results in experiment 1 are in line with the motivational priming theory, however, as I indicated earlier, this theory considers opposite effects for negative and positive stimuli. If the negative prime (painful stimulation) has an effect in the valence of the words, why are not influence in neutral words but if the positive?

Answer: We were not precise enough in our previous version. We now tried to give the pieces of results which are in accordance with the motivational priming theory more precisely (lines 338-344, 588-592). 

In addition, not all results are discuses, like why positive words were evaluated so much more negatively than negative ones? The motivational priming hypothesis indicates that in a negative context, the negative is exacerbated. In line with this, negative word under painful condition should be evaluate as more negative. However, the opposite effect was observed, where negative words are evaluated less negatively.

Answer: We fully agree. We had not presented a plausible discussion to positive adjectives. This has the reason that this was not our primary goal. We now tried to give a background for this difference (lines 636-657). 

Furthermore, suddenly, a study with PER is briefly described to try to explain the effect observed in the positive words. However, the explanation is unclear and departs from the methodology used. Should we have an N400 for this task? What process would be reflecting here?

Answer: We think that the argument of disagreement between expectation and occurrence of unexpected content is still valid. However, the study with N400 is far away and requires extensive explanation. As we now explain this point using the theory of predictive coding, we decided to remove this part of MS.

At the end of the first study, the authors indicated that not finding differences between words related to pain and negative ones was expected. However, this contradicts what was previously stated in the introduction section, lines 58 to 60.

Answer: We were not precise enough. There are different predictions with respect to emotional priming theory and network theory. We tried to clarify this throughout the MS.

In the case of experiment 2, there are not a discussion, just an explanation of the results. This part should be improved.

Answer: We changed and extended the discussion to Exp. 2 (lines 553-580).

---

## [Decision Letter · Decision Letter 1]

1 Feb 2021

PONE-D-20-29181R1

Influence of acute pain on valence rating of words

PLOS ONE

Dear Dr. Weiss,

Thank you for submitting your manuscript to PLOS ONE. I have received feedback from the two original reviewers. One of them recommended acceptance (pending some very minor changes). The other reviewer still have some concerns, mainly regarding the age of acquisition of the words, that should be addressed before publication. Please, try your best with the new version of the revised manuscript as i would like to make a final decision.

We look forward to receiving your revised manuscript.

Kind regards,

José A Hinojosa, Ph.D.

Academic Editor

PLOS ONE

Reviewers' comments:

Reviewer's Responses to Questions

**Comments to the Author**

1. If the authors have adequately addressed your comments raised in a previous round of review and you feel that this manuscript is now acceptable for publication, you may indicate that here to bypass the “Comments to the Author” section, enter your conflict of interest statement in the “Confidential to Editor” section, and submit your "Accept" recommendation.

Reviewer #1: (No Response)

Reviewer #2: All comments have been addressed

2. Is the manuscript technically sound, and do the data support the conclusions?

Reviewer #1: Partly

Reviewer #2: Yes

3. Has the statistical analysis been performed appropriately and rigorously? 

Reviewer #1: No

Reviewer #2: Yes

4. Have the authors made all data underlying the findings in their manuscript fully available?

Reviewer #1: Yes

Reviewer #2: Yes

5. Is the manuscript presented in an intelligible fashion and written in standard English?

Reviewer #1: Yes

Reviewer #2: Yes

6. Review Comments to the Author

Reviewer #1: The manuscript has been properly improved in many ways: a clearer rationale, the experimental procedure has been better described, a more informative data analysis and results, and the discussion goes more in-depth into the cognitive mechanisms that may be explaining the results.

However, I am afraid that I still cannot recommend the current version for publication because the two main concerns I raised in the first review have not been addressed.

First, the AoA of words is still unknown. If there were no AoA scores from previous studies for the words used in the experiments, the authors could have collected them through a rating study, which should not have been complicated given the low number of items to rate. In that case, if the mean AoA scores were similar by categories a possible confounding effect due to AoA of words would have been discarded; if the AoA scores were not similar by categories the speculation about a possible effect of AoA could have been state more exactly and based on data. In any case, I consider that including the omission of AoA scores as a limitation is an easy and not a sufficient solution.

On the other hand, a by-item analysis has not been carried out as I encouraged to perform. Without it we cannot rule out that the results are due to a few items that generate a strong effect. In other words, the effects are not generalizable to other (similar) words.

Reviewer #2: The manuscript has improved a lot by the changes made since the first submission. The authors have taken into account all comments and have responded appropriately to them. I would only like to suggest a few things:

In results section: when talking about significant differences found, it is convenient to always indicate in which direction these differences go in the text. Even though there is a figure or the means, it is useful that also indicated in the text. Like in lines 223, 264.

7. PLOS authors have the option to publish the peer review history of their article (what does this mean?). If published, this will include your full peer review and any attached files.

Reviewer #1: **Yes: **Miguel Á. Pérez-Sánchez

Reviewer #2: **Yes: **Irene Peláez Cordeiro

---

## [Author Response · Author response to Decision Letter 1]

2 Mar 2021

Responses to Journal Requirements and Reviewers

We thank the reviewers for taking time and effort to provide their constructive comments to improve the manuscript (Ms). We have addressed all the comments in a point-by-point reply. Our answers are indicated in blue font below. We provide a revised manuscript with track changes as well as an unmarked version of the revised manuscript. We report places of changes in this letter with respect to the marked-up copy of Ms. 

Reviewer 1

The manuscript has been properly improved in many ways: a clearer rationale, the experimental procedure has been better described, a more informative data analysis and results, and the discussion goes more in-depth into the cognitive mechanisms that may be explaining the results.

Thank you.

However, I am afraid that I still cannot recommend the current version for publication because the two main concerns I raised in the first review have not been addressed. First, the AoA of words is still unknown. If there were no AoA scores from previous studies for the words used in the experiments, the authors could have collected them through a rating study, which should not have been complicated given the low number of items to rate. In that case, if the mean AoA scores were similar by categories a possible confounding effect due to AoA of words would have been discarded; if the AoA scores were not similar by categories the speculation about a possible effect of AoA could have been state more exactly and based on data. In any case, I consider that including the omission of AoA scores as a limitation is an easy and not a sufficient solution.

Answer: 

We followed your advice and conducted a small rating study for the words used in our experiments to explore the AoA. Unfortunately, there is no German database containing the AoA of our adjectives. Birchenough, Davies, & Connelly (2017) delivered norms for over 3,200 German words, but only investigated the AoA of nouns and verbs. We used the same instructions and questionnaire as Birchenough, Davies, & Connelly (2017). We found similar mean AoA scores by categories. Details to this rating study are given in the Overall Discussion (lines 515-525). Furthermore, we adjusted the parts concerning AoA in Limitation and future directions (lines 546-558).

In addition, we searched the AoA norm by Birchenough, Davies, & Connelly (2017) for words similar to the ones we used in our study (with similar word stem) and found similar means to our rating study:

Birchenough, Davies, & Connelly (2017) our rating study

word Mean AoA word Mean AoA

Schimmel 5.9 schimmlig 6.0

stinken 3.6 stinkend 3.7

Dreck 4.0 verdreckt 3.3

gehen 2.9 gehend 2.8

blond 4.6 aschblond 4.4

Himmel 3.2 himmlisch 3.4

Zauber 5.5 bezaubernd 4.5

On the other hand, a by-item analysis has not been carried out as I encouraged to perform. Without it we cannot rule out that the results are due to a few items that generate a strong effect. In other words, the effects are not generalizable to other (similar) words.

Answer: 

We followed the advice and ran the analyses by item. We found a main effect for the factor Word and an interaction effect Category*Word. Post-hoc tests are presented in Table S2 in the supporting information. It seems that the words in the positive and neutral categories are rated more homogeneously concerning their overall valence in contrast to the pain-related and negative categories. However, we found no significant interaction between the factors Word and Prime. This indicates that the effect of the factors Prime and Category on valence ratings are not just due to a few items generating a strong effect. It rather shows a more general effect not just of individual words, but of word categories. We adjusted the affected parts in the Results and Discussion sections (lines 201/202, 212-217, 234-237, 317-320, 376, 391-398).

Reviewer 2

The manuscript has improved a lot by the changes made since the first submission. The authors have taken into account all comments and have responded appropriately to them. I would only like to suggest a few things: In results section: when talking about significant differences found, it is convenient to always indicate in which direction these differences go in the text. Even though there is a figure or the means, it is useful that also indicated in the text. Like in lines 223, 264.

Answer: 

We thank you for this advice. In accordance with your suggestion, we adjusted the results sections (lines 223/224, 228/229, 252-257, 272/273, 286, 406, 408, 415).

---

## [Decision Letter · Decision Letter 2]

5 Mar 2021

Influence of acute pain on valence rating of words

PONE-D-20-29181R2

Dear Dr. Weiss,

We’re pleased to inform you that your manuscript has been judged scientifically suitable for publication and will be formally accepted for publication once it meets all outstanding technical requirements.

Kind regards,

José A Hinojosa, Ph.D.

Academic Editor

PLOS ONE

Additional Editor Comments (optional):

Reviewers' comments:

Reviewer's Responses to Questions

**Comments to the Author**

1. If the authors have adequately addressed your comments raised in a previous round of review and you feel that this manuscript is now acceptable for publication, you may indicate that here to bypass the “Comments to the Author” section, enter your conflict of interest statement in the “Confidential to Editor” section, and submit your "Accept" recommendation.

Reviewer #1: All comments have been addressed

2. Is the manuscript technically sound, and do the data support the conclusions?

Reviewer #1: Yes

3. Has the statistical analysis been performed appropriately and rigorously? 

Reviewer #1: Yes

4. Have the authors made all data underlying the findings in their manuscript fully available?

Reviewer #1: Yes

5. Is the manuscript presented in an intelligible fashion and written in standard English?

Reviewer #1: Yes

6. Review Comments to the Author

Reviewer #1: The authors have considered all comments and have responded acceptably to them.

On the one hand, once the AoA of the words has been obtained, it does not seem to be a confounding effect due to that variable.

On the other hand, although the requested “item analysis” has not been performed in the standard way (i.e., considering items as the random factor), I think that a non-significant interaction between prime and word contributes to discard that the priming effects on valence ratings are not just due to a few items. BUT, I would only like to suggest that the result for the three-way interaction category*prime*word should be reported in order to remove any doubt about the effect of items on the critical interaction category*prime.

7. PLOS authors have the option to publish the peer review history of their article (what does this mean?). If published, this will include your full peer review and any attached files.

Reviewer #1: **Yes: **Miguel Á. Pérez-Sánchez

---

## [Editor Report · Acceptance letter]

9 Mar 2021

PONE-D-20-29181R2 

Influence of acute pain on valence rating of words 

Dear Dr. Weiss:

I'm pleased to inform you that your manuscript has been deemed suitable for publication in PLOS ONE. Congratulations! Your manuscript is now with our production department. 

Kind regards, 

on behalf of

Dr. José A Hinojosa 

Academic Editor

PLOS ONE